# MADGen: Minority Attribute Discovery in Text-to-Image Generative Models

## Abstract

Text-to-image diffusion models achieve impressive generation quality but also inherit and amplify biases from training data, resulting in biased coverage of semantic attributes. Prior work addresses this in two ways. Closed-set approaches mitigate biases in predefined fairness categories (e.g., gender, race), assuming socially salient minority attributes are known a priori. Open-set approaches frame the task as bias identification, highlighting majority attributes that dominate outputs. Both overlook a complementary task: uncovering minority features underrepresented in the data distribution (social, cultural, or stylistic) yet still encoded in model representations. We introduce MADGen, the first framework, to our knowledge, for discovering minority attributes in diffusion models. Our method leverages Matryoshka Sparse Autoencoders and introduces a minority metric that integrates neuron activation frequency with semantic distinctiveness, enabling the unsupervised identification of rare attributes. Specifically, MADGen identifies a set of neurons whose behavior can be directly interpreted through their top-activating images, which correspond to underrepresented semantic attributes in the model. Quantitative and qualitative experiments demonstrate that MADGen uncovers attributes beyond fairness categories, supports systematic auditing of architectures such as Stable Diffusion 1.5, 2, and XL, and enables amplification of minority attributes during generation.

## 1 Introduction

Text-to-image (T2I) diffusion models such as Stable Diffusion have revolutionized image generation by producing high-fidelity visuals from natural language prompts (Podell et al., 2023; Rombach et al., 2022). However, these models not only reflect biases from their training data (Luccioni et al., 2023; Perera & Patel, 2023), but can also amplify them during generation, reinforcing societal stereotypes and inequalities if left unaddressed(Seshadri et al., 2024). For instance, despite near-parity in the LAION-5B dataset for occupations like "teacher", generated samples remain heavily gender-skewed (Friedrich et al., 2023). Such disparities reduce semantic coverage and raise concerns about fairness, representation, and deployment in real-world settings.

Several approaches counteract biases in T2I generative models by rebalancing or diversifying their outputs (Chuang et al., 2023; Ni et al., 2023; Shen et al., 2024; Li et al., 2024). While effective for predefined categories like gender or race, they often overlook subtler underrepresentation such as physical traits, cultural symbols, or stylistic variations, essential for semantic diversity and faithful generation. Open-set bias detection (D'Incà et al., 2024) broadens auditing but mainly identifies the majority attributes and the surfaced attributes are largely dictated by inductive biases of external world models. Suppressing such majority features does not amplify the underrepresented ones, overlooking the critical task of identifying *minority attributes*: semantic factors encoded in the model's internal representations, but consistently underexpressed. Auditing remains incomplete without discovering them: it reveals what the model overproduces, but not what it neglects. Moreover, discovering minority attributes is vital for diversity and representation, counterbalancing imbalances and capturing cultural, stylistic nuances beyond fairness axes essential for realism and expressivity.

To address this gap, we introduce **MADGen**, the first framework for **Minority Attribute Discovery** in diffusion models without reliance on external language models. Rather than identifying all possible underrepresented attributes, MADGen targets those that are already encoded in the internal rep-

resentations of the model but are systematically underexpressed during generation. These attributes are not hallucinated or externally defined, but emerge directly from the learned feature space of the model. By shifting the focus from majority identification to the structured discovery of these suppressed features, MADGen enables more comprehensive auditing and a deeper understanding of representational gaps in generative models.

Identifying minority attributes requires access to the internal factors of variation learned by diffusion models. However, these representations are often entangled and uninterpretable. To address this, we require a mechanism that maps entangled internal representations into semantically meaningful, interpretable features – a role effectively realized by Sparse Autoencoders (SAEs) (Kim et al., 2025). We specifically adopt Matryoshka Sparse Autoencoders (MSAEs), which have shown strong interpretability in vision-language models via hierarchical semantic decomposition (Pach et al., 2025; Zaigrajew et al., 2025). We apply MSAE to diffusion activations and focus on the coarsest level, where neurons capture broad, high-level semantics. In contrast, finer layers often fragment concepts into narrow or stylistic variations, making them less reliable for discovering underrepresented attributes. Among coarse neurons, we identify minority-associated ones using a metric combining: (i) activation frequency, measuring how rarely a neuron fires, and (ii) semantic distinctiveness, measuring how dissimilar its top-activating samples are from the dataset distribution. This highlights neurons encoding rare yet semantically coherent concepts systematically underrepresented in outputs. Importantly, the interpretable structure of MSAEs enables direct visual interpretation through top-activating examples and spatial heatmaps, making explicit the attributes each neuron captures.

Empirically, we show that MADGen uncovers minority attributes across diverse prompts, extending beyond fairness-related categories. Beyond discovery, MADGen enables systematic auditing of diffusion architectures, including Stable Diffusion 1.5, 2, and XL, and demonstrates that identified attributes can be amplified at generation time through simple prompt revision.

The key contributions of this work are as follows: ❶ To the best of our knowledge, we introduce the first framework for minority attribute discovery in diffusion models, extending bias analysis from predefined fairness categories or majority-dominant features to the systematic identification of underrepresented attributes encoded in model representations. ❷ We propose a simple, yet effective, minority metric that combines neuron activation frequency with semantic distinctiveness, forming the basis of MADGen. ❸ We show that MADGen reveals attributes beyond fairness categories, enables auditing across multiple diffusion architectures (Stable Diffusion 1.5, 2, XL), and supports amplification via lightweight prompt interventions.

## 2 RELATED WORK

T2I generation has significantly advanced generative AI, enabling the creation of highly realistic images from textual prompts (Ho et al., 2020a; Ramesh et al., 2022; Rombach et al., 2022), but also inherit and amplify the biases in their training data (Cho et al., 2023; Luccioni et al., 2023).

**Bias Mitigation in Diffusion Models:** Recent work has increasingly focused on mitigating biases in diffusion models such as Stable Diffusion. Chuang et al. (2023) learn projection matrices on text embeddings aligned with fairness attributes, while methods such as Friedrich et al. (2023); Parihar et al. (2024) use classifier-free guidance to steer generations without retraining. A key limitation is their reliance on predefined minority attributes, assuming known targets for mitigation. In contrast, our approach is orthogonal: we seek to uncover minority attributes that are already encoded in the model but systematically underexpressed.

**Unknown bias identification:** Bias auditing in text-to-image models has shifted from mitigating predefined categories to identifying previously unknown biases. Open-set bias detection emphasizes uncovering such biases without relying on predefined labels. D'Incà et al. (2024) proposes a framework to automatically identify and quantify biases in generative models by leveraging large language models to suggest potential bias attributes, generating synthetic images, and applying visual question answering to rank the prevalence of these biases. However, such methods mainly surface majority attributes that dominate generations, revealing overrepresentation but not underrepresentation. We address this gap by shifting the focus from identifying dominant biases to discovering minority attributes that are suppressed.

**Interpretability with Sparse Autoencoders:** Sparse autoencoders have emerged as effective tools for interpreting generative models. Prior work demonstrates that intermediate activations in diffusion models can be mapped to human-interpretable concepts via sparse autoencoders (Kim et al., 2025; Surkov et al., 2025; Tinaz et al., 2025), enabling concept steering and suppression (Kim et al., 2025; Cywiński & Deja, 2025). Recent advances show that Matryoshka Sparse Autoencoders (MSAEs) offer hierarchical coarse-to-fine decompositions that improve interpretability in CLIP (Pach et al., 2025; Zaigrajew et al., 2025). While prior efforts focus on interpretability and control, we extend the use of MSAEs to minority attribute discovery. Our framework introduces a novel minority metric that identifies features that are both underrepresented and semantically coherent, broadening the role of sparse autoencoders to systematic bias auditing in diffusion models.

## 3 PRELIMINARIES

**Diffusion Models:** Diffusion models (Sohl-Dickstein et al., 2015; Ho et al., 2020b; Song & Ermon, 2019) synthesize data by learning to reverse a forward process that progressively adds Gaussian noise to a clean sample $\mathbf{x}_0 \sim p_{\text{data}}$ according to a variance schedule, yielding $\mathbf{x}_T \sim \mathcal{N}(\mathbf{0}, \mathbf{I})$ as $t \to T$. A neural network $\boldsymbol{\epsilon}_\theta(\mathbf{x}_t, t)$ learns to predict the added noise, defining a denoising transition at each step. At inference, generation starts from noise and recursively applies the learned reverse process to produce a clean sample. Text-to-image models such as Stable Diffusion (Rombach et al., 2022) extend this framework by operating in a compressed latent space where they condition on text embeddings from a language encoder, thereby aligning generated images with natural language.

**Sparse Autoencoders (SAEs):** Sparse autoencoders aim to decompose input representations $\mathbf{r} \in \mathbb{R}^n$ into a set of latent features $\mathbf{z} = \{z_1, \ldots, z_d\} \in \mathbb{R}^d$ that are both overcomplete ($d \gg n$) and sparse, thereby encouraging interpretability and disentanglement of concepts. The encoder-decoder architecture is defined as:

$$\mathbf{z} = \text{ReLU}(W_{\text{enc}}(\mathbf{r} - \mathbf{b}_{\text{pre}}) + \mathbf{b}_{\text{enc}}), \qquad \hat{\mathbf{r}} = W_{\text{dec}}\mathbf{z} + \mathbf{b}_{\text{pre}},$$

where $W_{\text{enc}} \in \mathbb{R}^{n \times d}$, $W_{\text{dec}} \in \mathbb{R}^{d \times n}$, and $\mathbf{b}_{\text{enc}} \in \mathbb{R}^d, \mathbf{b}_{\text{pre}} \in \mathbb{R}^n$ are learnable parameters. The model is trained to minimize the reconstruction loss $\mathcal{L}_{\text{SAE}} = \|\mathbf{r} - \hat{\mathbf{r}}\|_2^2$, while enforcing sparsity on $\mathbf{z}$. Sparsity is imposed either through $\ell_1$ penalties on $\mathbf{z}$ (Bricken et al., 2023), which can cause activation shrinkage (Rajamanoharan et al., 2024), or by hard selection of the top-$k$ coordinates per input (Gao et al., 2025), which enforces exact sparsity but fixes the number of active units. BatchTopK (Bussmann et al., 2025) modifies this by flattening all activations in a batch into a single vector and retaining the largest $k \times B$ entries (for batch size $B$), allowing the number of active features to vary across samples while maintaining a global sparsity constraints.

**Matryoshka Sparse Autoencoders (MSAEs):** MSAEs extend SAEs by training under multiple sparsity constraints at once, following the idea of Matryoshka representation learning (Kusupati et al., 2022). Instead of selecting a single sparsity level $k$, the model applies a family of Top-$k$ operators with increasing levels $\{k_1, k_2, \ldots, k_f\}$, where $k_1 < k_2 < \cdots < k_f = d$, forming a nested budget: $k_1$ active neurons at the first level, then $(k_2 - k_1)$ more at the next, and so forth, up to $d$. For an input $\mathbf{r}$, the encoder produces multiple sparse codes and reconstructions for each level as follows:

$$\mathbf{z}^{(k_i)} = \text{ReLU}\big(\text{Top}_{k_i}(W_{\text{enc}}(\mathbf{r} - \mathbf{b}_{\text{pre}}) + \mathbf{b}_{\text{enc}})\big), \qquad \hat{\mathbf{r}}^{(k_i)} = W_{\text{dec}}\mathbf{z}^{(k_i)} + \mathbf{b}_{\text{pre}}.$$

The training objective aggregates reconstruction losses across all levels,

$$\mathcal{L}_{\text{MSAE}} = \sum_{i=1}^{n} \alpha_i \|\mathbf{r} - \hat{\mathbf{r}}^{(k_i)}\|_2^2, \tag{1}$$

with coefficients $\alpha_i$ weighting the contribution of each sparsity level. At inference, any $k_i$ can be probed to reveal features at varying granularities. This design produces a hierarchical representation, with coarse levels capturing broad semantics and finer levels encoding detailed attributes.

## 4 METHODOLOGY

We propose **MADGen**, a framework for minority attribute discovery in text-to-image diffusion models. An overview of the framework is illustrated in Figure 1.

Figure 1: Overview of **MADGen**. Diffusion representations ($\mathbf{h}$) are decomposed by MSAE into interpretable features ($\mathbf{z}$). A minority score ($s$), combining rarity and distinctiveness, ranks neurons or features to reveal *minority attributes*. Minority concepts are identified at the coarsest MSAE level (e.g., female doctor, doctor in suit), where size reflects activation frequency (smaller size = less frequent) and color denotes neuron identity.

## 4.1 PROBLEM FORMULATION

Let $\mathcal{A} = \{a_1, a_2, \ldots, a_m\}$ denote the set of semantic attributes that may be expressed in the outputs of a conditional generative model, where $m \geq 2$. For instance, $\mathcal{A}$ could include {male, female, urban background, rural background, dark skin tone, ...}. A conditional generative model is defined as $G : (\boldsymbol{\xi}, \mathbf{c}) \mapsto \mathbf{x}$, where $\boldsymbol{\xi} \sim \mathcal{N}(\mathbf{0}, \mathbf{I})$ is a latent variable, $\mathbf{c} \in \mathcal{C}$ is an external condition (*e.g.*, a text prompt), and $\mathbf{x}$ is the generated output. The model induces a conditional probability distribution over attribute values:

$$P_G(a_i \mid \mathbf{c}) = \Pr[A(\mathbf{x}) = a_i \mid \mathbf{c}], \quad i = 1, \ldots, m.$$

where $A(\mathbf{x})$ denotes the attribute value associated with the generated sample $\mathbf{x}$.

**Definition 1** (**Generative Bias**). A model $G$ exhibits *generative bias* (Ferrara, 2024; Huang & Huang, 2025) with respect to attribute set $\mathcal{A}$ under condition $\mathbf{c}$ if there exist $i \neq j$ such that

$$P_G(a_i \mid \mathbf{c}) \neq P_G(a_j \mid \mathbf{c}).$$

That is, the model assigns uneven probabilities to attribute values under identical conditions.

**Definition 2** (**Minority Attribute**). For a tolerance parameter $\epsilon > 0$, an attribute $a_j \in \mathcal{A}$ is a *minority attribute* under condition $\mathbf{c}$ if

$$0 < P_G(a_j \mid \mathbf{c}) \leq \min_{a_i \in \mathcal{A} \setminus \{a_j\}} P_G(a_i \mid \mathbf{c}) + \epsilon$$

This definition implies two conditions: (1) attributes with $P_G(a_j \mid \mathbf{c}) = 0$ are excluded, as they are not represented in the model's latent space; (2) minority attributes are defined with respect to the model's generative distribution rather than the raw training data, identifying features that are internally encoded but suppressed in outputs.

**Objective:** The discovery of minority attributes requires a mechanism that meets the following criteria: (1) It exposes the set of internal semantic concepts encoded by the generative model $G$. Formally, let $\mathbf{h}$ denote the representations extracted from $G$. We aim to utilize a feature decomposition operator $M$ that maps the representations into a set of sparse latent features $M(\mathbf{h}) = \mathbf{z} = \{z_1, \ldots, z_d\} \in \mathbb{R}^d$. Throughout, we use the term *neuron* to refer to individual latent feature $z_i$ in the sparse representation $\mathbf{z}$, each of which may capture a distinct semantic feature. (2) It assigns a quantitative score reflecting the degree of underrepresentation of each feature. Specifically, we define a scoring function $s$ that assigns each latent feature $z_i$ a value $s(z_i) \in [0, 1]$, which yields the score vector $s(\mathbf{z}) = (s(z_1), \ldots, s(z_d)) \in [0, 1]^d$. Each $s(z_i)$ captures both the rarity of feature $z_i$ under condition $\mathbf{c}$ and its semantic distinctiveness relative to other features in $\mathbf{z}$.

This motivates MADGen, which employs MSAEs to decompose the representations $\mathbf{h}$ into semantically meaningful features $\mathbf{z}$, and introduces a novel *minority score* $s(\mathbf{z})$ that integrates feature rarity with semantic distinctiveness. Together, these components enable the unsupervised discovery of minority attributes directly from the internal representations of diffusion models.

## 4.2 Minority Attribute Discovery

**Feature Decomposition from Diffusion Representations:** To expose the internal concepts encoded by diffusion models, we train MSAE on intermediate representations extracted during reverse sampling. Given a T2I diffusion model $G$ and a prompt $\mathbf{c}$, we extract bottleneck representations $\mathbf{h}_t \in \mathbb{R}^{h \times w \times n}$ at each denoising step $t$. These representations are inherently interpretable (Kwon et al., 2023), and Kim et al. (2025) has shown that SAEs trained on them reveal high-level features. Following Cywiński & Deja (2025), we treat each spatial location in $\mathbf{h}_t$ as an $n$-dimensional training example, disregarding spatial coordinates. These vectors are then used to train a MSAE using Equation 1, yielding a hierarchy of sparse codes $\mathbf{z}^{(k_i)}$ that capture semantic structure at varying levels of granularity from broad concepts at coarse levels ($k_1$) to finer details at deeper levels($k_f$).

For minority attribute discovery, we perform inference with MSAE by collecting representation-image pairs $\mathcal{D}_c = \{ (\mathbf{h}_t^{(j)}, \mathbf{x}^{(j)}) \}_{j=1}^N$, where $\mathbf{h}_t^{(j)} \in \mathbb{R}^{h \times w \times n}$ denotes bottleneck representation at a fixed denoising step $t$, and $\mathbf{x}^{(j)}$ is the corresponding generated image for a prompt $c$. In practice, we use the final timestep, where semantic information is most fully expressed. For simplicity, we omit both the sample index $j$ and the timestep index $t$, and write $(\mathbf{h}, \mathbf{x}) \in \mathcal{D}_c$ for an arbitrary pair. Each tensor $\mathbf{h}$ is flattened into $h \times w$ feature vectors, which are individually passed through the MSAE encoder following the training setup. For each MSAE neuron $z_i$, with $i = 1, \ldots, d$ corresponding to a sparse latent feature, we define its activation on $\mathbf{h}$ as $z_i(\mathbf{h})$, obtained by averaging activations across spatial positions. These per-neuron activations form the basis for computing the minority score.

**Minority Score:** To quantify the degree to which each neuron encodes a minority attribute, we introduce the *Minority Score*, which balances two complementary signals: rarity of activation and semantic distinctiveness. Let $(\mathbf{h}, \mathbf{x}) \in \mathcal{D}_c$ be an diffusion representation-image pair, and $z_i(\mathbf{h})$ the activation of MSAE neuron $z_i$ as previously defined. We define the activation frequency as the proportion of samples where the neuron is active (i.e., has nonzero activation):

$$\nu_i = \frac{|\{(\mathbf{h}, \mathbf{x}) \in \mathcal{D}_c : z_i(\mathbf{h}) > 0\}|}{|\mathcal{D}_c|}. \tag{2}$$

This metric directly measures how often the neuron participates across the dataset $\mathcal{D}_c$, with rarer features corresponding to lower $\nu_i$. While activation frequency identifies neurons that fire rarely, rarity alone is insufficient: some neurons may activate sparsely but correspond to noisy, low-level fluctuations that are not meaningful for interpretation. To address this, we evaluate the semantic distinctiveness of each neuron by comparing its activation-weighted CLIP centroid to the global dataset centroid. Let $\text{CLIP}(\mathbf{x})$ denote the CLIP embedding of image $\mathbf{x}$. The centroid $\mu_i$ for neuron $z_i$, and the global centroid $\mu_{\mathcal{D}_c}$, are computed as:

$$\mu_i = \frac{\sum_{(\mathbf{h}, \mathbf{x}) \in \mathcal{D}_c} z_i(\mathbf{h}) \text{CLIP}(\mathbf{x})}{\sum_{(\mathbf{h}, \mathbf{x}) \in \mathcal{D}_c} z_i(\mathbf{h})}, \qquad \mu_{\mathcal{D}_c} = \frac{1}{|\mathcal{D}_c|} \sum_{(\mathbf{h}, \mathbf{x}) \in \mathcal{D}_c} \text{CLIP}(\mathbf{x}). \tag{3}$$

Semantic distinctiveness $d_i$ is then defined as the cosine distance between the two centroids. This metric ensures that the neuron centroid $\mu_i$ is dominated by its top-activating images, since activation strengths directly weight their contribution. In contrast, the global dataset centroid $\mu_{\mathcal{D}_c}$ represents the average semantics of the entire set of images in $\mathcal{D}_c$, dominated by the majority distribution. The resulting distance $d_i$ thus measures how much the concept encoded by a neuron diverges from dominant patterns in the data. Both $d_i$ and $\nu_i$ are min–max normalized to $[0, 1]$ for comparability. Finally, *Minority Score* is defined as:

$$s(\mathbf{z}) = \mathbf{d} \odot (\mathbf{1} - \boldsymbol{\nu}) \tag{4}$$

where $\mathbf{d} = (d_1, \ldots, d_d)$, $\boldsymbol{\nu} = (\nu_1, \ldots, \nu_d)$. This formulation assigns a high score to neurons that are both rarely active (low $\nu_i$) and semantically distinct (high $d_i$) relative to the majority distribution. Intuitively, neurons with larger $s(z_i)$ are more likely to encode minority attributes, since they capture concepts that occur infrequently yet deviate substantially from dominant patterns. Conversely, neurons with the lowest minority scores do not necessarily correspond to dominant attributes, since low values can also arise from noisy or undistinctive activations, as we explain in detail in Section A.5.1. Although the Minority Score can be computed across all MSAE neurons, we focus on the coarsest level $z(k_1)$, which captures broad, interpretable semantics. By restricting analysis to

the top-$k_1$ codes, we prioritize high-level structure over low-level noise, leveraging the hierarchical design of MSAEs to expose global attributes more clearly than standard SAEs.

Minority concepts often appear redundantly across multiple neurons with similar activation patterns. To obtain a compact and diverse set, we use the neuron centroid $\mu_i$ (Eq. 3) as a representative of each neuron's semantics. Redundancy is assessed via pairwise cosine distances between centroids, which quantify similarity between neurons. We then iterate over neurons in descending order of *Minority Score*, retaining the current neuron in the final set and removing all others within a small fixed distance. This threshold, treated as a hyperparameter, controls semantic redundancy. To further ensure that the final set contains minority neurons, we restrict analysis to those above the 90th percentile of the score distribution. This cutoff is relative rather than absolute: it surfaces the most suppressed and distinctive neurons within a domain, making the final group more likely to correspond to minority attributes. Hyperparameters were selected through sweeps and guided by heuristics, and validated using interpretability checks, ensuring that the surviving neurons capture coherent, human-readable concepts. The resulting set therefore contains distinct, interpretable, minority neurons ranked by underrepresentation. For interpretability, we visualize top-activating images with MSAE heatmaps and provide human-readable annotations via LLMs.

## 5 EXPERIMENTS

We conduct qualitative and quantitative evaluations of the minority attributes discovered by MAD-Gen, then audit bias across multiple diffusion architectures (Stable Diffusion 1.5, 2, and XL) using these attributes. Finally, we show that minority attributes are not only discoverable but can also be systematically amplified during generation. We also include additional experiments such as user study and various ablation studies on different components of MADGen in Section A.5.

### 5.1 MINORITY ATTRIBUTES DISCOVERY

**Datasets:** We perform minority attribute discovery on WinoBias (Zhao et al., 2018) and COCO prompts (Lin et al., 2015) to ground our analysis in datasets that capture complementary aspects of bias and diversity. WinoBias provides controlled, occupation-based prompts that are widely used in fairness evaluations, making it well-suited for testing whether MADGen can surface socially salient minority attributes such as gender imbalances in profession-related generations. In contrast, COCO offers a broad and diverse distribution of everyday scenes, and contexts, allowing us to evaluate whether MADGen can uncover underrepresented attributes beyond fairness categories, such as cultural artifacts or stylistic cues.

**Experimental setting:** We perform minority attribute discovery on Stable Diffusion v1.4 (Rombach et al., 2022). For each prompt in WinoBias and COCO, we generate images and hook their bottleneck representations. These representations are used to train the MSAE, after which we identify the underrepresented attributes in SDV1.4 as dicussed in Section 4. For interpretability, the minority neurons are annotated using GPT-4o. Additional experimental details are provided in Section A.3.

We assess the effectiveness of MADGen using two complementary metrics. ❶**Likelihood:** This metric quantifies how probable the discovered minority attributes are under the generative model's distribution. For each neuron identified as encoding a minority attribute, we evaluate the likelihoods of its top-activating images and compare them against randomly sampled images from SD under the same prompts. Minority attributes are expected to lie in lower-density regions of the distribution, resulting in lower likelihood values. ❷ **Attribute presence:** This metric measures how often the minority attributes discovered by MADGen appear in images generated from SD. We use the language annotations obtained for these attributes as described in Section 4, and for each generated image, evaluate whether the annotated attribute is detected by the LLaMA-4 Scout model. Lower presence values indicate stronger underrepresentation. We compare MADGen against Open-Bias (D'Incà et al., 2024), which targets majority attributes. As no prior method addresses minority attribute discovery, this provides a complementary perspective: while open-set methods expose what the model overproduces, MADGen uncovers what it systematically suppresses. Additional details on evaluation metrics are provided in Section A.4.

**Quantitative results:** We evaluate MADGen using average likelihood and attribute presence metrics to assess its effectiveness in uncovering minority attributes in diffusion models. Table 1 com-

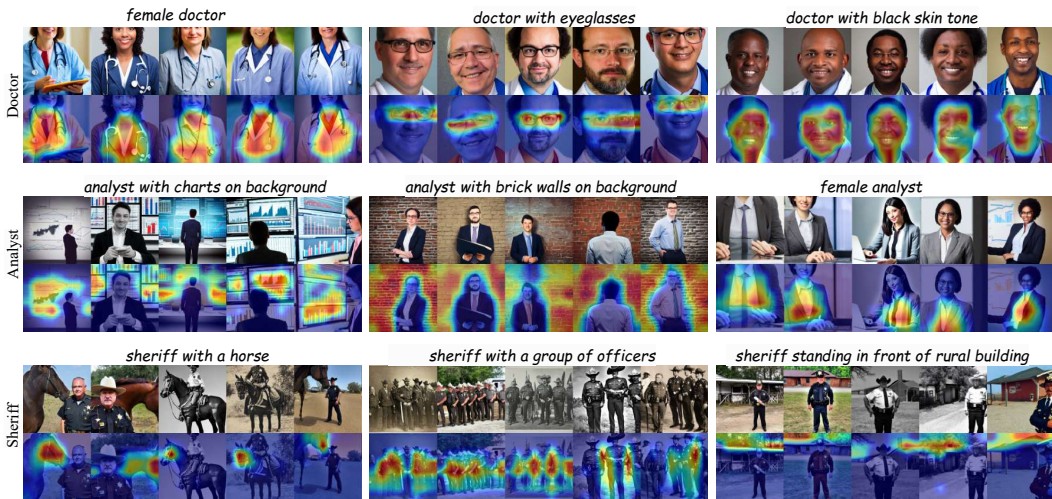

Figure 2: Top-activating images and MSAE activation heatmaps for minority neurons discovered by MADGen across three WinoBias prompts. **Top:** "Doctor." **Middle:** "Analyst." **Bottom:** "Sheriff." The label above each image shows its language-based annotation.

pares average likelihoods for random Stable Diffusion samples with those associated with minority neurons identified by MADGen. We observe that MADGen produces significantly lower values, indicating that the discovered attributes lie in low-density regions of the distribution. Although these features are encoded in the representation space, the images they generate have reduced likelihoods and are thus less likely to appear, supporting our definition of minority attributes as systematically underrepresented yet nonzero in probability.

Table 1: Average likelihood measured in bits per dimension.

Table 2: Attribute presence for majority (Open-Bias) and minority (MADGen).

| Approach | WinoBias ($\downarrow$) | COCO ($\downarrow$) |
|---|---|---|
| Stable Diffusion | 2.394 | 2.378 |
| MADGen | **2.371** | **2.334** |

| Approach | WinoBias ($\downarrow$) | COCO ($\downarrow$) |
|---|---|---|
| OpenBias | 0.941 | 0.933 |
| MADGen | **0.256** | **0.265** |

To assess how frequently the discovered attributes are expressed during generation, we evaluate MADGen using the attribute presence metric. Table 2 compares the average presence of attributes identified by MADGen against those discovered by OpenBias. We observe that attributes from OpenBias occur with high frequency, while those surfaced by MADGen appear substantially less often. This large divergence from majority attributes provides direct evidence that MADGen is isolating features that are underrepresented, consistent with our definition of minority attributes. The results also highlight the complementary focus of the two methods: OpenBias captures dominant modes of the distribution, whereas MADGen reveals rare modes that the model seldom expresses.

**Qualitative results:** To evaluate the qualitative impact of MADGen, we visualize MSAE neuron activations and their corresponding GPT-4o annotations. Figure 2 shows results for WinoBias prompts, while Figure 3 presents results for COCO. On WinoBias, MADGen uncovers both socially salient minority attributes (e.g., female, black skin tone) and non-fairness concepts (e.g., analysts in front of brick walls, sheriffs in rural settings). On COCO, it identifies rare stylistic and contextual features, such as black-and-white portraits or side-view trains with motion blur. These results demonstrate that MADGen generalizes beyond fairness attributes and reveals a broad range of underrepresented semantics in diffusion models. Additional qualitative results are provided in Section A.7.

## 5.2 CROSS-MODEL ANALYSIS OF MINORITY ATTRIBUTES WITH MADGEN

In this section, we use MADGen as a systematic auditing framework to compare minority attribute representations across diffusion architectures. Focusing on the profession of *Doctor*, we apply MADGen independently to SD v1.4, v2.1, and XL to identify minority neurons and annotate their

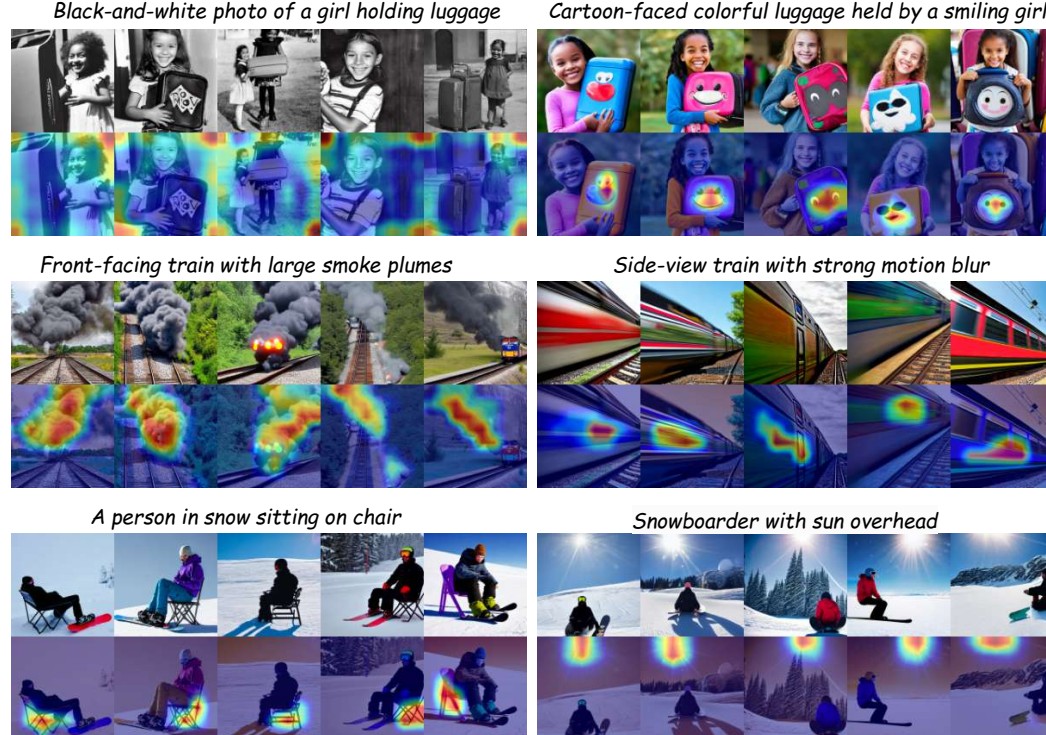

Figure 3: Top-activating images and MSAE activation heatmaps for minority neurons discovered by MADGen across COCO prompts. **Top:** "A girl smiling as she holds a piece of luggage." **Middle:** "A train going down a track at full speed." **Bottom:** "A person sitting in the snow with snowboard on." The label above each image shows its language-based annotation.

semantics. We then collect the unique set of discovered attribute annotations across all models to form a unified attribute vocabulary. For each attribute in this set, we measure its presence in each model by generating 1000 samples from the corresponding model and evaluating attribute presence, as described in Section A.4.2. This enables us to analyze whether an attribute identified as a minority in one architecture remains underrepresented or amplified in others, as model design evolves.

The results are summarized in Figure 4. Demographic attributes show the strongest change: SD v1.4 heavily underrepresents female and black doctors, partially corrected in SD v2.1, while SDXL introduces new imbalances, such as overemphasis on traits like beards. Stylistic minorities follow the reverse trend—attributes like sepia tone, vintage clothing, and outdoor settings appear in SD v1.4 but are nearly absent in SDXL, which favors standardized, formal portrayals. Contextual minorities shift most dramatically: diverse backdrops (e.g., bookshelves, hospital corridors) in early models are replaced by a dominant clinical office setting in SDXL, indicating contextual homogenization. In contrast, gestural attributes such as arms crossed or sitting remain stable across versions, suggesting pose features are robustly preserved.

By surfacing minority attributes and enabling their tracking across model families, MADGen reveals how architectural changes and scaling can shift underrepresentation rather than resolve it. Gains in demographic balance may trade off with stylistic or contextual diversity, highlighting the need for auditing of all forms of underrepresentation beyond narrow fairness categories.

## 5.3 AMPLIFICATION OF DISCOVERED MINORITY ATTRIBUTES DURING GENERATION

We investigate whether minority attributes identified by MADGen can be used to amplify underrepresented modes in the generative distribution and mitigate representational bias. Using annotations surfaced by MADGen, we apply prompt revision guided by LLaMA-4 Scout, which injects minority descriptors into the input text. While our framework is agnostic to mitigation strategies (Chuang et al., 2023; Friedrich et al., 2023), we adopt prompt revision as a lightweight intervention to test whether reintroducing minority concepts improve generation.

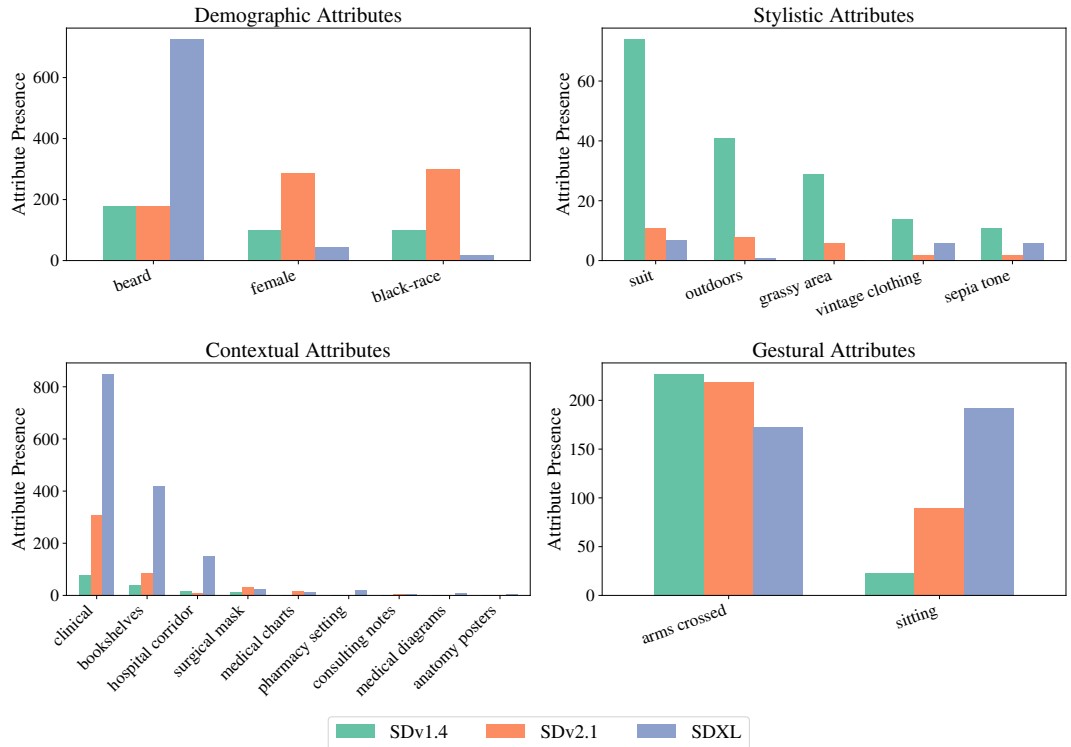

Figure 4: Category-level presence of minority attributes in images of doctors across SD versions. Demographic, stylistic, contextual, and gestural attributes reveal distinct representational shifts.

We evaluate this on COCO and Wino-Bias prompts by uniformly sampling images under two conditions: the original base prompt and a revised prompt incorporating the minority attribute. For each case, we measure (i) likelihood under the base prompt, (ii) deviation from a uniform attribute distribution, and (iii) CLIP alignment with the original prompt. Additional details on the metrics are provided in Section A.4.3.

Table 3: Comparison of WinoBias generations with and without prompt revision using MADGen attributes.

| Approach | Likelihood (↓) | Deviation (↓) | CLIP Alignment (↑) |
|---|---|---|---|
| Stable Diffusion | 2.394 | 0.377 | **20.00** |
| MADGen | **2.343** | **0.199** | 19.57 |

Table 3 reports results averaged over 36 WinoBias prompts. Minority-guided generations yield lower likelihoods, consistent with sampling from low-density regions. Deviation from uniformity drops by nearly half, indicating improved balance across attributes without collapsing to majority modes. Additionally, CLIP alignment remains comparable, showing that semantic fidelity is preserved. Additional results on COCO are provided in Section A.5.2. Overall, MADGen also serves as a mechanism for amplifying unknown underrepresented attributes beyond fixed fairness groups.

## 6 CONCLUSIONS

We propose MADGen, a framework for minority attribute discovery in diffusion models that combines Matryoshka Sparse Autoencoders with a novel minority score to identify features encoded in latent representations but underrepresented at generation. Unlike prior work focused on fixed fairness categories or majority trends, MADGen uncovers rare, semantically meaningful attributes directly from internal activations. Through quantitative and qualitative analyses, we show that MADGen reveals fairness, stylistic, and cultural minorities, enables cross-model audits across Stable Diffusion variants, and facilitates targeted amplification via prompt revision. By grounding discovery in model representations, MADGen complements LLM-based bias tools and lays the foundation for hybrid auditing frameworks that bridge external priors with internal model dynamics.

## 7 ETHICS STATEMENT

This work focuses on the discovery of minority attributes in diffusion models with the goal of improving fairness, representation, and diversity in generative systems. The identification of minority attributes is intended solely for auditing and research purposes, to highlight underrepresented features and ensure that they are appropriately considered in model development and deployment. We acknowledge that some socially relevant attributes may not be identified as minorities in our framework, as the results depend on the performance and limitations of sparse autoencoders. We therefore caution that our method should be seen as a tool to surface overlooked attributes, rather than a definitive or exhaustive catalog of underrepresentation. In addition, we conduct a user study to evaluate the perceived validity of surfaced attributes. All participants were informed about the purpose of the study, provided consent before participation, and were free to withdraw at any time without penalty. No personally identifiable information was collected, and responses were analyzed in aggregate to protect privacy. Care was taken to avoid exposing participants to harmful or offensive content.

## 8 REPRODUCIBILITY STATEMENT

We have taken considerable care to ensure the reproducibility of our results by providing the necessary details throughout the manuscript. Section 5 of the main paper describes the datasets and experimental settings used for minority attribute discovery. Section A.3 of the appendix presents additional experimental details, including the hyperparameters required for reproducibility, and Section A.4 of the appendix outlines further evaluation procedures. Our implementation will be released publicly upon acceptance.

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

# A  APPENDIX

In the primary text of our submission, we introduce MADGen, a framework for uncovering minority attributes encoded in the internal representations of diffusion models. To preserve clarity and conciseness in the main paper, we provide an extensive appendix that complements the core manuscript. The appendix includes additional experiments, detailed implementation protocols, broader qualitative analyses, and deeper ablations that could not fit within the page limits. Together, these materials extend the discussion in the main text by offering a fuller view of our methodology, empirical validation, and implications.

## A.1  DISCUSSION AND FUTURE WORKS

A promising future direction is the integration of LLM-based and representation-based approaches for minority attribute discovery. While MADGen surfaces attributes that are internally encoded but suppressed during generation, LLM-based tools offer a complementary perspective by drawing on external world knowledge to hypothesize socially salient or semantically expected minorities.

Since no existing method explicitly targets minority attribute discovery via LLMs, we perform a preliminary study by adapting OpenBias, originally designed for majority (overrepresented) feature identification for this purpose. While not validated for minority detection, we repurpose its language-based pipeline to surface expected but underrepresented attributes (e.g., non-binary cashiers, Hispanic teachers) and compare them against those discovered by MADGen (e.g., doctors in vintage attire, teachers outdoors). Adapting OpenBias for minority discovery yields attributes with near-zero presence in generated samples (0.021 for WinoBias, 0.058 for COCO), reflecting concepts that the model fails to encode altogether. This highlights the complementarity of the two approaches: LLM-based methods surface expectation-driven minorities aligned with human priors, while MADGen reveals underexpressed concepts that are internally encoded. Together, they provide a broader view of underrepresentation, motivating future work on unified frameworks that combine external semantic priors with internal representation-grounded discovery.

## A.2  LIMITATIONS

While MADGen provides a systematic framework for discovering minority attributes in diffusion models, it is not without limitations. First, our approach only captures attributes that are encoded in the model's internal representations, meaning that socially salient fairness categories entirely absent from the representation space (e.g., non-binary identities, certain cultural groups) will not be surfaced. This limits the scope of fairness auditing to what the model has already learned, rather than what society may expect. Second, since MADGen relies on sparse autoencoders for interpretability, the quality and granularity of discovered attributes depend heavily on what the autoencoder itself captures. Attributes may be fragmented, merged, or overlooked depending on the sparsity budget and training dynamics, and alternative architectures could yield different results. Third, even among the minority neurons that are identified, some may be noisy or ambiguous. In the absence of ground truth annotations for minority attributes, it is difficult to rigorously quantify coverage and precision of the discovered neurons. These limitations should be considered when applying MADGen, particularly in fairness auditing and downstream interventions. Nonetheless, they do not diminish the standalone utility of the approach in surfacing underrepresented concepts, but rather highlight opportunities to further strengthen it when combined with complementary tools.

## A.3  EXPERIMENTAL DETAILS

This section provides a detailed account of the experimental setup, including datasets, training procedures, and hyperparameter choices, to ensure that our results can be reliably reproduced.

### A.3.1  DISCOVERY OF MINORITY ATTRIBUTES

We utilize 36 WinoBias professions and 100 COCO prompts to investigate the effectiveness of MADGen. For WinoBias, prompts are constructed in the form "A photo of a *profession*", where the profession is drawn the benchmark, following prior text-to-image generation frameworks. For COCO, we directly use the original prompt captions without modification. For each prompt

in WinoBias and COCO, we generate 5000 images together with their bottleneck representations of size $1280 \times 8 \times 8$ across all timesteps. Each spatial location at each timestep is treated as an independent sample, yielding a 1280-dimensional vector that serves as input to the MSAE. This setup is adopted Cywiński & Deja (2025). We train MSAE using a sparse latent feature space of size $1280 \times 16$, employing two sparsity levels: a coarse level with $k_1 = 2048$ neurons, and the remaining neurons allocated to the fine level. We adopt the official implementation provided by Bussmann et al. (2025) for training, with the following hyperparameters: 5 training epochs, an effective batch size of $4,096$, a learning rate of $3 \times 10^{-4}$, and an auxiliary penalty weight of $\frac{1}{32}$.

Following training, we generate 5000 images along with their bottleneck representations of size $1280 \times 8 \times 8$ at timestep=49. The number of timesteps during sampling from diffusion models is set to 50. To compute semantic distinctiveness, we use OpenAI's CLIP ViT-L/14 model to extract embeddings for the top-activating images of each neuron. For interpretability, we use the OpenAI GPT-4o model to annotate the minority neurons. During the pruning, we fix a cosine distance threshold of 0.003. To further ensure semantic relevance, we restrict the set to neurons whose minority scores exceed the 90th percentile.

We now provide the prompt that we used to annotate our neurons using GPT-4o:

```
You are a JSON-only generator. You are not allowed to explain anything,
    write markdown, or comment. Only return a single valid JSON object.
    You have to analyze a specific neuron from a sparse autoencoder
    trained on profession-related generated images. Each neuron activates
     in response to specific, consistent visual features that appear
    across its top-activating images, as confirmed by corresponding
    heatmaps.

- The input prompt always follows the form: "a photo of a <profession>"

- You are provided with:
  - Top-activating images (first row): strongest activations for this
      neuron
  - Heatmaps (second row): regions most responsible for the neuron
      activation

- Your job is:
  1. To carefully observe the top-activating images and heatmaps, and
      identify visually consistent attributes that correlate with the
      neuron activation.
  2. To generate a modified version of the base prompt that includes
      these attributes naturally and precisely.
  3. To output a flat list of non-redundant keywords capturing only the
      consistent attributes.

- Strict requirements:

  - The identified attributes must be:
    - Clearly and consistently visible across the top-activating images
    - Highlighted or partially supported by the heatmap attention
    - Not already implied by the base prompt
    - Not a core object/tool expected for the profession

{{
    "neuron_id": "{neuron_id}",
    "input prompt": "{base_prompt}",
    "identified_attribute": "<short, precise description of all
        consistent visual attributes across the images>",
    "suggested_prompt": "<the base prompt modified to include these
        attributes naturally>",
    "keywords": ["<keyword_1>", "<keyword_2>", "..."]
}}

### Example Outputs
```

```
Input Prompt: "a photo of a doctor"

Example 1:
{{
    "neuron_id": "2041",
    "input prompt": "a photo of a doctor",
    "identified_attribute": "female doctor",
    "suggested_prompt": "a photo of a female doctor",
    "keywords": ["female"]
}}
```

### A.3.2 CROSS-MODEL ANALYSIS OF MINORITY ATTRIBUTES

We adopt the same experimental setting used for SD v1.4 to identify minority attributes in SD v2.1 and SDXL, ensuring consistency across model comparisons.

### A.4 EVALUATION DETAILS

In this section, we provide additional evaluation details that we utilized to investigate the effectiveness of MADGen in identifying underrepresented attributes.

### A.4.1 DISCOVERY OF MINORITY ATTRIBUTES

We evaluate the effectiveness of MADGen in discovering minority attributes using two complementary metrics: Likelihood and Attribute presence.

**Likelihood.** For each neuron identified as encoding a minority attribute, we select its top-10 activating images and compute their exact log-likelihoods conditioned on the prompt using the PF-ODE estimator of Song et al. (2020), implemented in `unified_metric`[1]. The likelihoods are aggregated across images for each prompt. As a baseline, we report the same statistics on 500 randomly sampled images from Stable Diffusion per prompt. We then average across prompts to obtain the average likelihood. Lower values indicate that minority attributes occupy low-density regions of the generative distribution, consistent with their underrepresentation.

**Attribute Presence.** This metric quantifies how often the discovered minority attributes appear in generated samples. For this evaluation, we rely on the minority attribute annotations produced with GPT-4o, as described in Section 4. For each generated image, the LLaMA-4 Scout model is provided with the image and a candidate minority attribute, and we record whether the attribute is detected using the prompt *"Is the attribute present in the image?"*. This process is repeated for all minority attributes under consideration, and the results are aggregated across images for each prompt, We then average across prompts to obtain the average attribute presence. Lower values indicate stronger underrepresentation.

For each profession in Winobias, we consider all minority attributes discovered by MADGen. Likelihood is computed on the top-10 activating images per neuron and compared against 500 random samples per profession. Attribute presence is measured using the same full set of discovered attributes. For COCO, we restrict to the top-10 highest-scoring minority neurons per prompt, as the remaining neurons add little beyond near-duplicate or low-confidence attributes, unlike WinoBias where many distinct variants emerge. Likelihood is again computed on the top-10 activating images per neuron and compared against 500 random samples per prompt. For attribute presence, we consider the same top-10 neurons per prompt.

### A.4.2 CROSS-MODEL ANALYSIS OF MINORITY ATTRIBUTES WITH MADGEN

We evaluate the effectiveness of utilizing MADGen to compare minority attribute representations across diffusion architectures using a variant of the Attribute presence metric by quantifying how often the discovered minority attributes appear in generated samples. We use MADGen to first

---

[1]https://github.com/unified-metric/unified_metric

identify minority neurons and annotate their associated semantics for SD v1.4, v2.1, and XL. We then aggregate all unique discovered attributes into a unified vocabulary.

For each model, we then generate 1000 images conditioned on the prompt. For every candidate attribute in the unified vocabulary, the LLaMA-4 Scout model is provided with the generated image and asked whether the attribute is present using the prompt *"Is the attribute present in the image?"*, and we record whether the attribute is detected. We repeat this process across all attributes. This produces an occurrence count that allows us to compare how frequently each attribute manifests across models.

### A.4.3 Amplification of Minority attributes

We evaluate the effectiveness of amplifying the minority attributes discovered by MADGen during generation using three metrics: likelihood, discrepancy from a uniform distribution, and CLIP alignment. In the amplification setting, we generate images for each attribute under two conditions: (i) the base prompt, and (ii) a revised prompt augmented with the discovered minority attribute. For each image, one of these two conditions is randomly selected, ensuring a balanced mixture of base and revised generations across the evaluation set.

**Likelihood.** Likelihood is computed as described above. For each attribute, we compute the average log-likelihood across all generated images and then aggregate across all attributes under consideration. Lower values indicate that the amplified samples remain in low-density regions of the distribution, consistent with their underrepresented nature.

**Discrepancy.** Discrepancy measures how evenly images are distributed between the base and revised prompts. Intuitively, if amplification is successful, the distribution of samples across attributes should approach uniformity. Formally, this metric is defined analogously to fairness discrepancy measures in prior works (Parihar et al., 2024), where smaller values indicate better balance between base and revised generations.

**CLIP Alignment.** Finally, we measure the semantic alignment between the generated images and their corresponding textual prompts using CLIP embeddings. Higher alignment scores indicate that amplification strengthens the consistency between the intended minority attribute and the visual output, without sacrificing prompt fidelity.

For both WinoBias and COCO prompts, we evaluate amplification using the top-5 minority attributes per prompt, generating 100 images for each prompt under this setting.

### A.5 Additional experiments

In this section, we provide additional experiments that we performed to investigate the effectiveness of MADGen in identifying underrepresented attributes.

### A.5.1 Why Low Minority Scores Do Not Capture Majority Features

While the *Minority Score* is well suited to isolate underrepresented features (low frequency, high distinctiveness), we investigate whether neurons with the lowest scores might instead recover dominant attributes. To this end, we utilize MADGen to identify minority neurons from the intermediate representations of diffusion models for the prompt *"A photo of a Doctor"*. We then examine the bottom-ranked neurons from our trained MSAE at coarse sparsity levels, where semantic content is most fully expressed. We visualized their activations and corresponding heatmaps, using Figure 5 to illustrate representative cases.

Empirically, we observe that the dominant features appear duplicated, fragmented, or spread across many neurons whereas minority features emerge as sharp, semantically coherent units. This arises because the majority attributes correspond to broad, high-variance regions of the diffusion representation space, which are distributed across multiple correlated directions rather than concentrated along a single axis. As a result, the neurons with the lowest Minority Scores remain diffuse and unreliable for interpretation, often localizing to narrow, fine-grained patches in the heatmaps rather than capturing the attribute holistically. This variation persists even under strong sparsity, forcing the

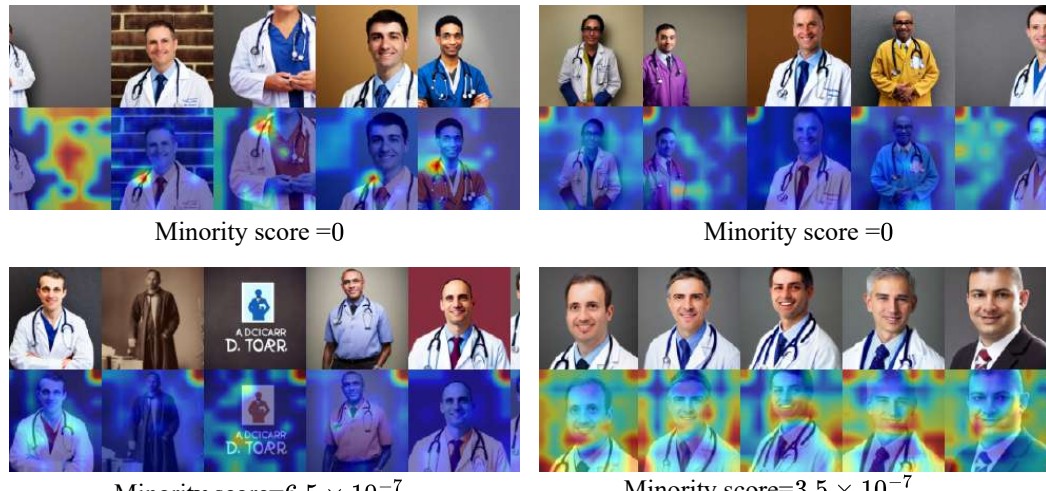

Minority score =0         Minority score =0

Minority score=$6.5 \times 10^{-7}$      Minority score=$3.5 \times 10^{-7}$

Figure 5: Low-scored (majority) neurons identified by MADGen for the prompt *"A photo of a doctor"*. Unlike minority neurons, their activations and heatmaps are diffuse, fragmented, and fail to capture coherent semantic attributes.

model to distribute reconstruction responsibility across several correlated neurons. By contrast, minority concepts occupy compact, lower-variance regions with little redundancy, enabling the MSAE to assign a single neuron to capture the full feature without incurring large reconstruction penalties. Consequently, low scores do not provide clean access to dominant ones and hence, the framework is expressly tailored for the discovery of minority attributes, and not for the recovery of interpretable majority features.

### A.5.2 ADDITIONAL RESULTS ON AMPLIFICATION OF MINORITY ATTRIBUTES ON COCO

To complement our evaluation in Section 5.3, we assess how the minority attributes discovered by MADGen can be utilized to amplify their generation. We consider the top-5 minority attributes identified by MADGen for 100 COCO prompts. For each attribute, we generate images with and without revised prompts uniformly and compute the likelihood under the base prompt distribution. We also compute attribute deviation from uniformity and CLIP alignment. As in the main paper, likelihood values are compared against a baseline of 500 randomly sampled images for each COCO prompt. The results are summarized in Table 4.

Table 4: Comparison of COCO generations with and without prompt revision using MADGen attributes.

| Approach | Likelihood ($\downarrow$) | Deviation ($\downarrow$) | CLIP Alignment ($\uparrow$) |
|---|---|---|---|
| Stable Diffusion | 2.378 | 0.51 | **27.06** |
| MADGen | **2.322** | **0.29** | 25.65 |

As observed in Table 4, minority-guided generations on COCO yield lower likelihoods than random Stable Diffusion samples, indicating that they lie in lower-density regions of the model's distribution. At the same time, deviation from uniformity decreases substantially, suggesting improved balance across minority attributes. Finally, we note a modest drop in CLIP alignment compared to Stable Diffusion. This is expected, since alignment is measured against the original base prompts, while our images are generated using revised prompts that explicitly inject minority descriptors. As a result, generations are less likely to match the base prompt verbatim, leading to slightly lower scores. The effect is more pronounced on COCO, where prompts are open-domain and more diverse. Importantly, this limitation arises because we use simple prompt revision as the mitigation strategy. More advanced interventions that leverage our discovered minority attributes, such as controlled decoding or targeted steering could overcome this issue and maintain both diversity and prompt fi-

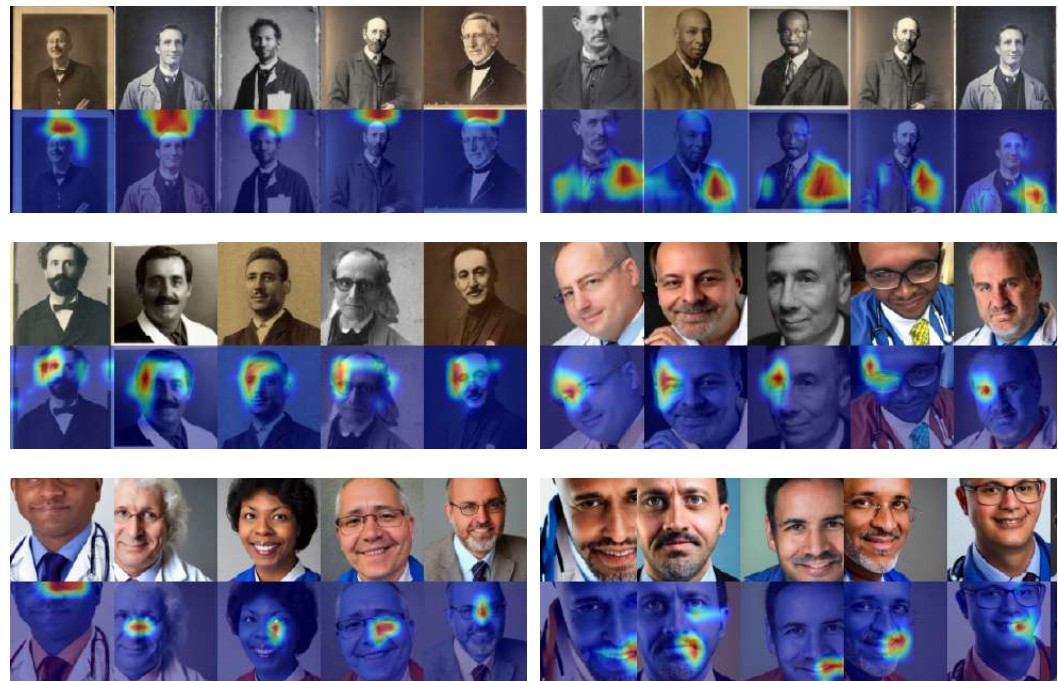

Figure 6: Visualizations of top-activating samples and corresponding heatmaps for top minority neurons identified using a standard SAE for MADGen. Each pair shows the top-activating images and the neuron's activations. While the neurons capture meaningful localized features (e.g., facial details or specific contextual elements), they frequently fragment broader concepts across multiple neurons, illustrating reduced interpretability compared to MSAE.

delity. Our aim here is only to demonstrate that the attributes identified by MADGen can indeed be used to mitigate underrepresentation, rather than to optimize the mitigation strategy itself. Together with the WinoBias results in the main paper, these findings demonstrate that MADGen can amplify underrepresented features across both controlled and real-world settings.

### A.5.3 USER STUDY

To assess whether minority attributes identified by MADGen are perceptible to humans and systematically underrepresented, we conducted a user study with 25 participants. We selected five professions Analyst, CEO, Doctor, Salesperson, Sheriff and, for each, the top-6 minority attributes discovered by MADGen. Each participant was shown a grid of 10 images per profession (sampled from Stable Diffusion v1.4) and asked, for each attribute, *"How many of the images in this grid contain this attribute?"* Responses ranged from 0 to 10. Although MADGen identifies underrepresented attributes directly from intermediate model representations, we consider their corresponding generated images as the perceptual ground truth in this user study. This design ensures that the human judgments reflect whether the attributes surfaced by MADGen are indeed visible and underexpressed in the outputs of the generative model.

For each attribute, we aggregated across participants to compute the mean presence, i.e., the average number of images (out of 10) in which participants reported the attribute. A mean of 0 indicates the attribute never appears, while a mean of 10 indicates the attribute appears in all images for all participants. We then averaged these values across attributes within each profession, and the results are summarized in Table 5.

As observed in Table 5, on average, attributes appear in fewer than 3 images per grid, far from the maximum of 10. Notably, professions like CEO exhibit particularly low presence ($< 1$ image per grid), reflecting severe underrepresentation, while Sheriff shows higher but still incomplete coverage (2–3 images per grid). These findings validate MADGen's ability to surface meaningful

Table 5: Average mean presence of minority attributes across professions (max = 10).

| Profession | Avg. Mean Presence ($\downarrow$) |
|---|---|
| Analyst | 1.39 |
| CEO | 0.70 |
| Doctor | 1.18 |
| Salesperson | 1.29 |
| Sheriff | 2.60 |

but suppressed attributes beyond chance noise, enabling more comprehensive auditing of generative models.

### A.5.4 MADGEN USING VANILLA SAE

While our primary experiments utilize MSAEs for hierarchical semantic decomposition, we also conduct a parallel analysis using standard SAEs to evaluate how the choice of decomposition method affects the discovery of minority attributes. We apply MADGen to intermediate representations of diffusion models for the prompt *"A photo of a Doctor"*. In this setting, rather than using MSAE, we employ a vanilla SAE to identify neurons associated with underrepresented attributes. We visualize several of the top minority neurons discovered by our approach, with the results summarized in Figure 6.

To evaluate the role of the decomposition method, we repeated our analysis using a standard Sparse Autoencoder (SAE) in place of the Matryoshka SAE. As shown in Figure 6, the SAE was still able to surface minority attributes under our *Minority Score*, with neurons capturing features such as eyeglasses and contextual backdrops. However, consistent with observations in Bussmann et al. (2025), SAE representations were considerably more fragmented, often distributing a single concept across multiple neurons. For example, we identified cases where one neuron responded primarily to the right eye and another to the left eye; although both were flagged as minority neurons due to their selective activations, they did not reflect genuine semantic minorities but rather over-fragmented features. This fragmentation reduced semantic coherence, making annotation less straightforward and limiting interpretability. Overall, these results demonstrate that while SAEs can uncover minority attributes, their tendency to fragment concepts across multiple neurons can be misleading, as neurons may appear to represent minorities while in fact capturing only partial or redundant features. In contrast, MSAEs mitigate this issue by providing hierarchical control over granularity and emphasizing global semantic features, enabling more faithful and interpretable discovery of genuinely underrepresented attributes.

### A.5.5 ABLATION ON MSAE COARSE LEVEL SIZE FOR MADGEN

In this section, we perform an ablation on the number of neurons in the coarse sparsity level in MSAE which we utilize for MADGen to understand its effect on the minority attribute discovery. We ablate the MSAE coarse sparsity level size $k_1 \in \{1280, 2048, 10240\}$ while fixing the sparse feature dimension to 20480 (input representation size = 1280, expansion factor = 16), and evaluate on the prompt *"A photo of a doctor"*. As shown in Table 6, mean likelihood remains essentially stable, while attribute presence increases at larger $k_1$. Qualitatively, we observed that some demographic attributes (e.g., Black skin tone) are missed at $k_1{=}1280$ but captured at $k_1{=}2048$. Balancing coverage against fragmentation, we adopt $k_1{=}2048$ as the default, which reliably recovers key demographics without the over-fragmentation seen at very large $k_1$.

### A.5.6 ABLATION ON DIFFERENT COMPONENTS IN MINORITY SCORE

In this section, we analyze the effectiveness of different components, such as the Activation frequency of MSAE and the distinctiveness, to compute the *Minority Score* as discussed in Section 4. The evaluations are done on the prompt *"A photo of a doctor"*. We first analyze the case where neurons are ranked only by frequency, and the visualization of the least frequently activated neurons are provided in Figure 7. While these appear high under frequency-only ranking, their top-activating images are noisy and uninterpretable, and lack semantic consistent meaning. Conversely, when we

| Number of neurons in $k_1$ | Likelihood | Attribute presence |
|---|---|---|
| 1280 | 2.568 | 0.15 |
| 2048 | 2.574 | 0.15 |
| 10240 | 2.577 | 0.20 |

Table 6: Ablation on the number of neurons in coarse sparsity level $k_1$ in MSAE for MADGen. Likelihood and attribute presence are computed as in our evaluation protocol.

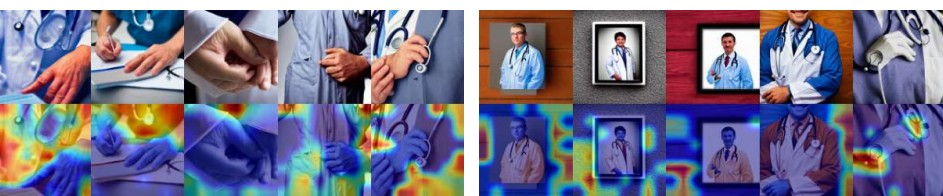

Figure 7: (a) Frequency-only minority neuron identification: Neuron with (Left) Frequency 0.01, CLIP distance 0.56, and (Right) Frequency 0.06, CLIP distance 0.51. These are the least frequently activated neurons, but appear noisy and uninterpretable.

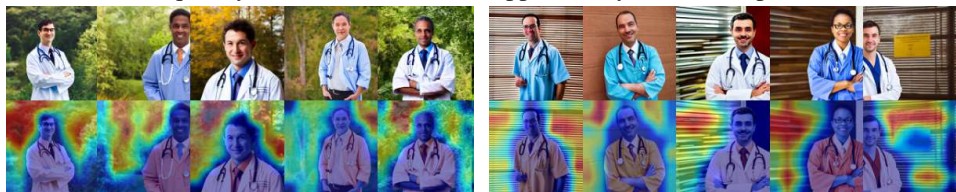

Figure 8: (b) Semantic distinctiveness-only minority neuron identification: Neuron with (Left) Frequency 0.05, CLIP distance 0.06, and (Right) Frequency 0.12, CLIP distance 0.12. These are the least semantically distinct from the global centroid, but are actually low-frequency minority neurons.

rank solely by semantic distinctiveness as observed in Figure 8, we encounter the opposite failure mode: neurons with very low CLIP distance to the global centroid are deemed non-distinct, even though they activate rarely and thus genuinely represent minority patterns. Such neurons would never be selected under distinctiveness-only ranking, despite being true minorities. To overcome these complementary limitations, we adopt a combined criterion that considers both frequency and semantic distinctiveness, enabling robust identification of neurons that are both rare and semantically coherent.

### A.6 USE OF LARGE LANGUAGE MODELS

We used large language models (LLMs) in two ways during this work. First, LLMs were employed to polish the writing of the manuscript, improving clarity and readability without altering the technical content. Second, we used LLMs to annotate the minority attributes identified by MADGen, automating the process of generating human-readable descriptions. This reduced the annotation burden on human evaluators while ensuring that discovered attributes could be systematically analyzed at scale.

### A.7 QUALITATIVE RESULTS

In this section, we present qualitative results illustrating the minority neurons identified by MADGen across different datasets and prompts. Figures 9 and 10 show minority neurons corresponding to different Winobias professions from the representations of SD v1.4. Figure 11 displays minority neurons obtained for a set of diverse COCO prompts in SD v1.4. Finally, Figures 12 and 13 present the minority neurons identified for the prompt *"A photo of a Doctor"* in SD v2.1 and SDXL, respectively. For each neuron, we strongly activating images along with their corresponding activation

heatmaps, which highlight the visual features. Since MADGen surfaces several minority neurons, we present only a representative subset in the figures for clarity and illustration.

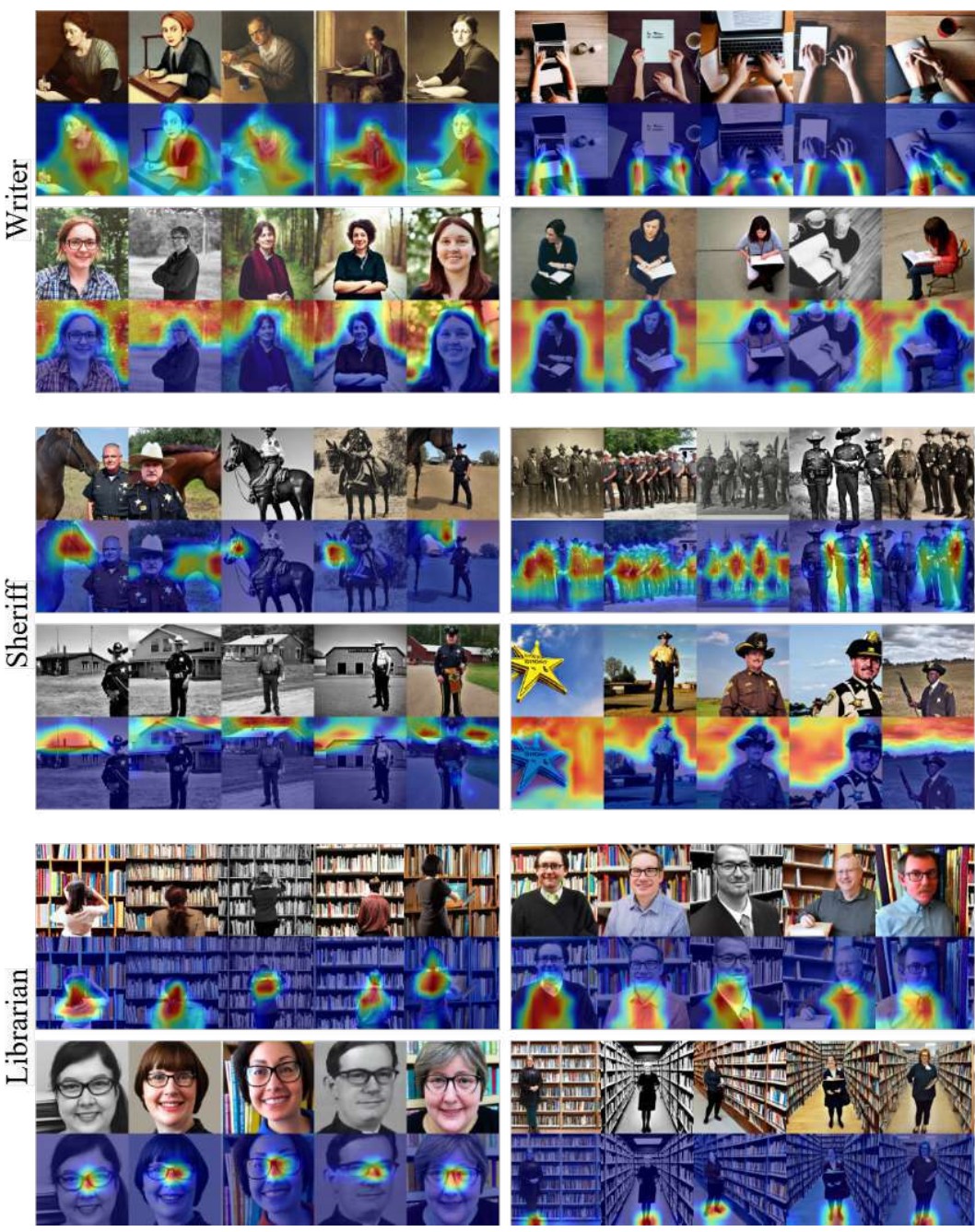

Figure 9: Visualization of 4 minority neurons obtained by MADGen for different Winobias professions. For each neuron, we show the top five activating images and their corresponding activation heatmaps.

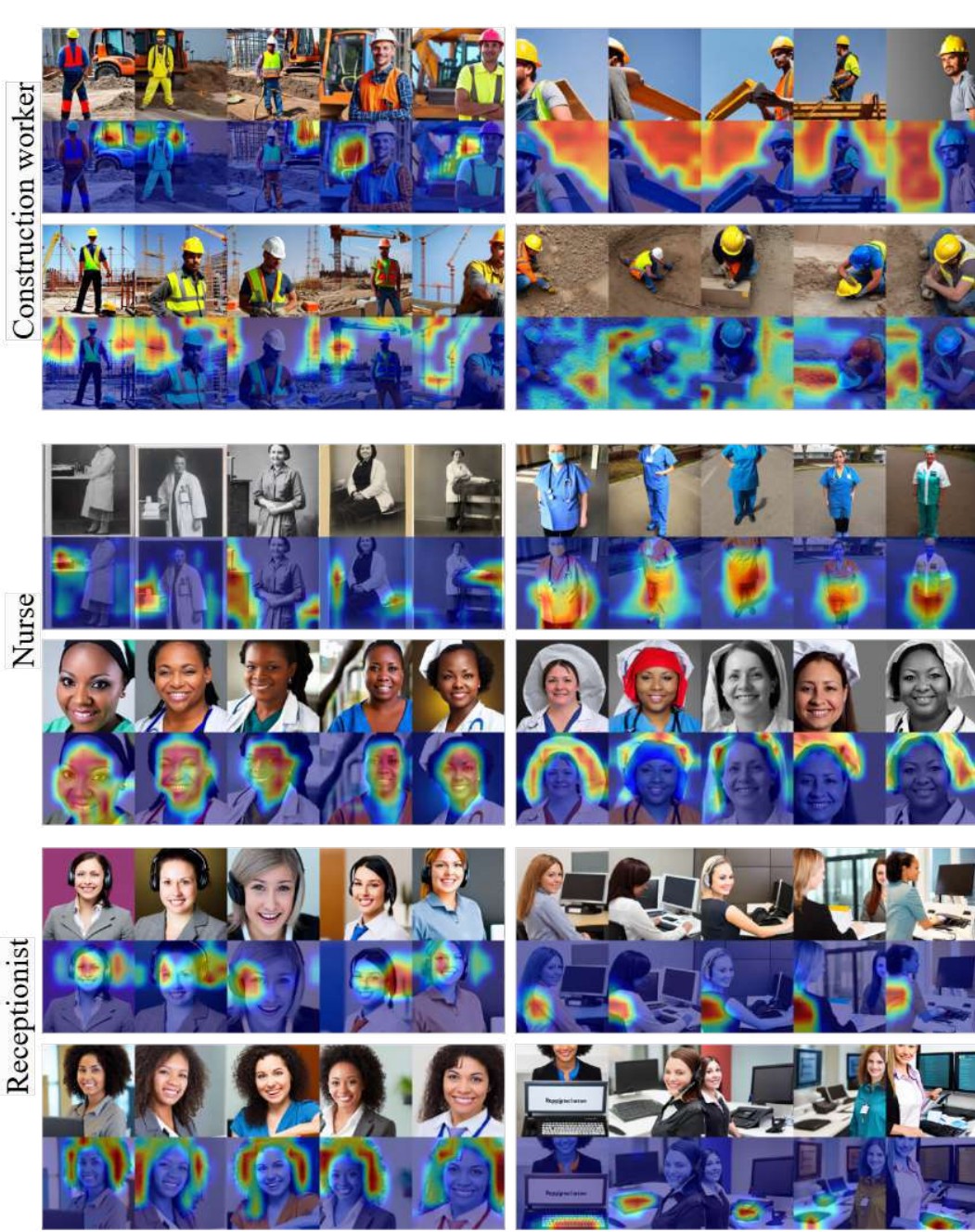

Figure 10: Visualization of 4 minority neurons obtained by MADGen for different Winobias prompts. For each neuron, we show the top five activating images and their corresponding activation heatmaps.

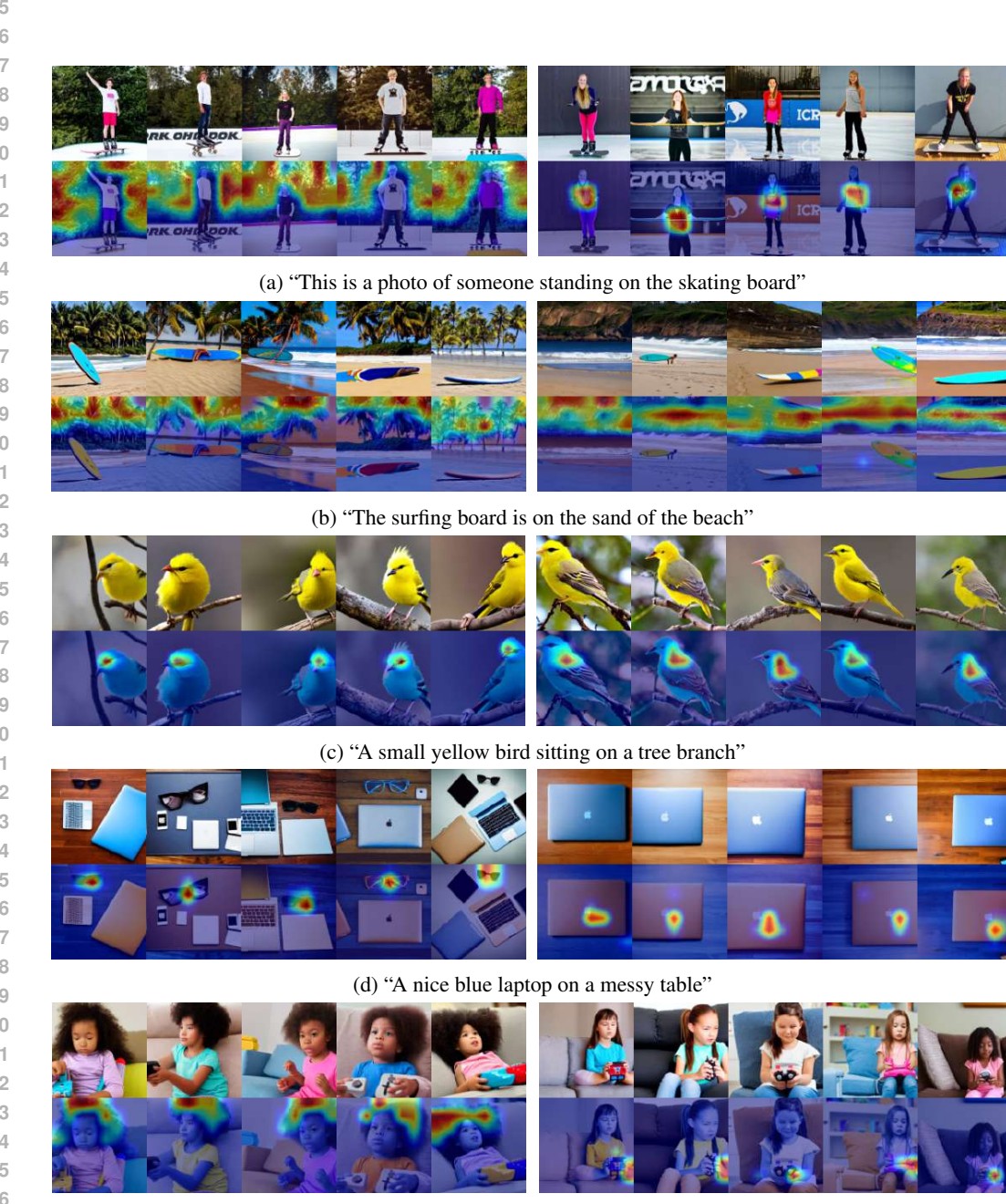

(a) "This is a photo of someone standing on the skating board"

(b) "The surfing board is on the sand of the beach"

(c) "A small yellow bird sitting on a tree branch"

(d) "A nice blue laptop on a messy table"

(e) "A little girl sitting on the couch playing"

Figure 11: Visualization of 2 minority neurons obtained by MADGen for different COCO prompts. For each neuron, we show the top five activating images and their corresponding activation heatmaps.

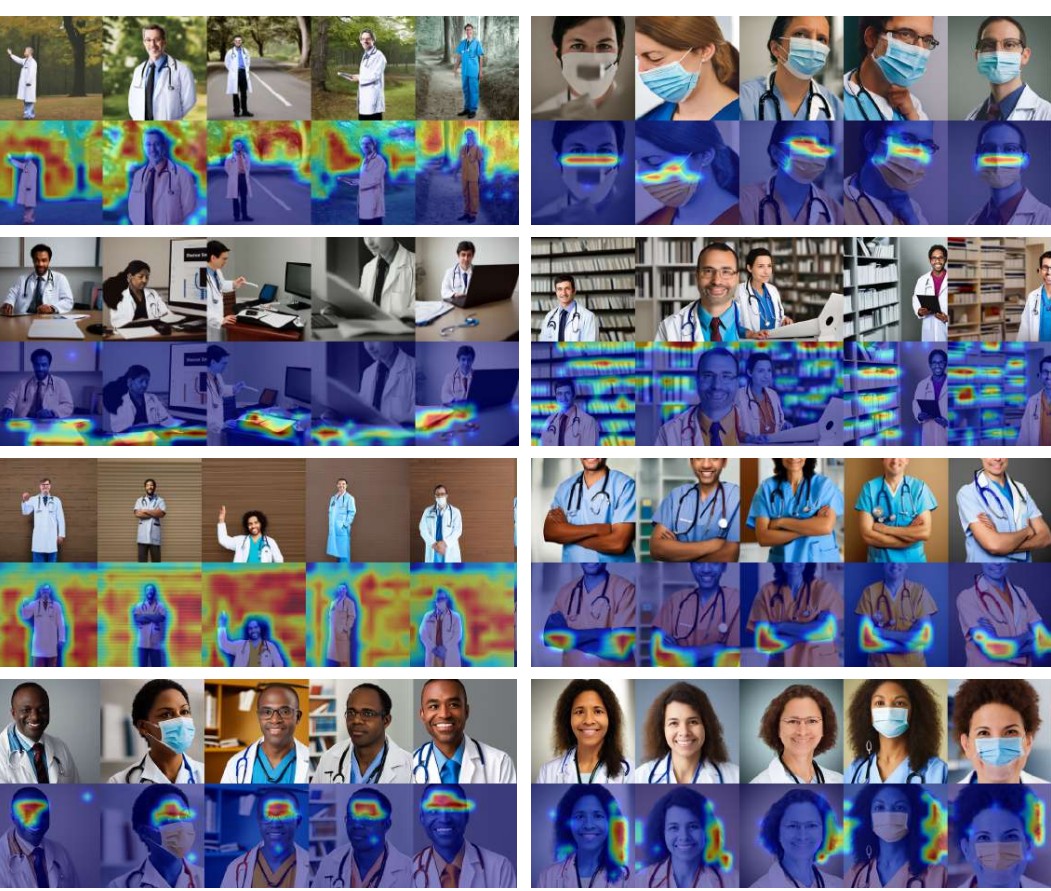

Figure 12: Visualization of 8 minority neurons identified by MADGen for the prompt *"A photo of a Doctor"* in SD v2.1. For each neuron, we show the top five activating images and their corresponding activation heatmaps.

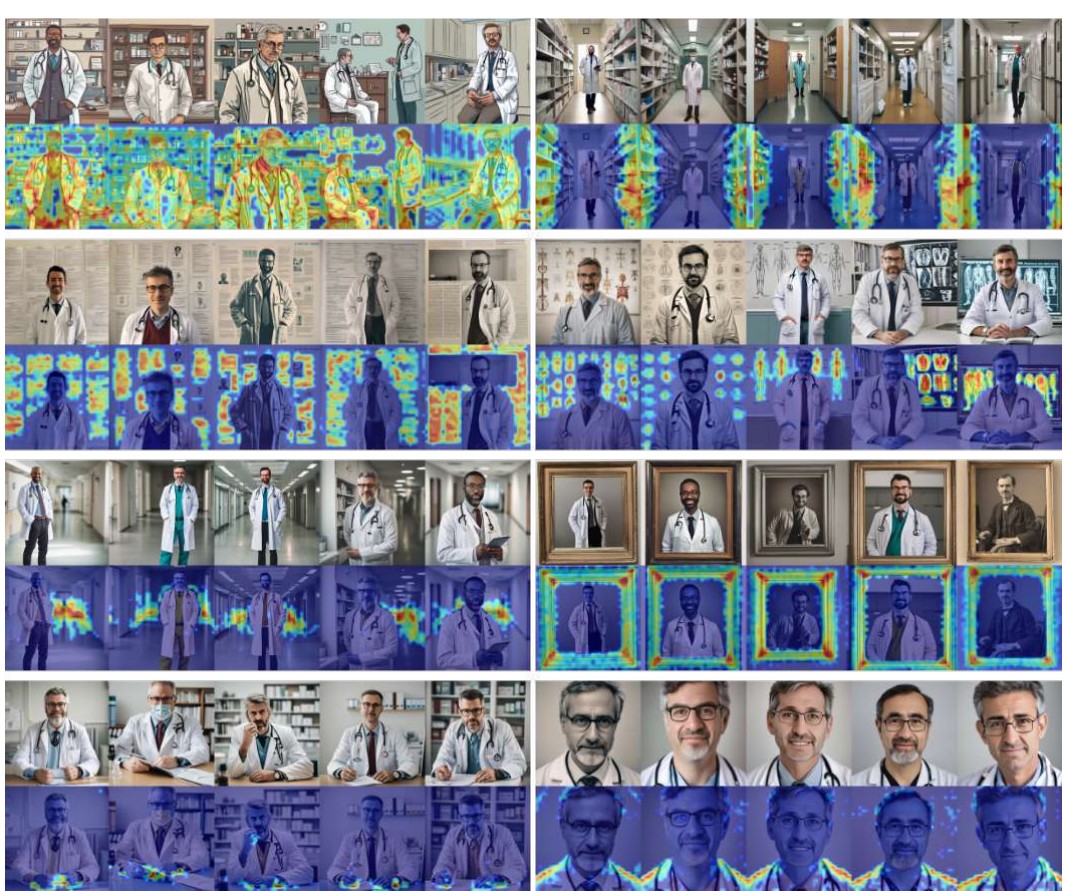

Figure 13: Visualization of top 8 minority neurons identified by MADGen for the prompt *"A photo of a Doctor"* in SDXL. For each neuron, we show the top five activating images and their corresponding activation heatmaps.

