# OpenReview forum: "MADGen:Minority Attribute Discovery in Text-to-Image Generative Models"
_ICLR.cc/2026/Conference — ICLR 2026 Conference Withdrawn Submission_

### Official Review · Reviewer_ehoC · 2025-10-30

**Soundness:** 1
**Presentation:** 2
**Contribution:** 1
**Rating:** 2
**Confidence:** 4

**Summary:**

This paper introduces MADGen, a framework intended to discover "minority attributes" in text-to-image diffusion models. These attributes are defined as semantic concepts that are encoded in the model's internal representations but are systematically underrepresented in the generated outputs. The method trains a Matryoshka Sparse Autoencoder (MSAE) on the U-Net's intermediate activations to find interpretable features (neurons). It then proposes a "Minority Score," $s(z) = d \odot (1-\nu)$, which combines semantic distinctiveness ($d$, based on CLIP) and activation rarity ($1-\nu$), to rank these neurons. The authors claim this framework can audit models and demonstrate its use across Stable Diffusion v1.4, v2.1, and XL.

**Strengths:**

The paper's only significant strength is identifying and articulating the important, unsolved problem of "minority attribute discovery," distinguishing it from the more common tasks of closed-set mitigation or majority bias detection.

**Weaknesses:**

1. The method's core premise that low activation frequency ($\nu_i$) indicates systematically suppressed attributes is an unvalidated heuristic.
2. The paper's primary qualitative "proof" is unconvincing and self-contradictory. The authors use fragmented heatmaps in Figure 5 to dismiss low-scoring neurons as "non-coherent". However, a supposedly "successful" high-scoring neuron in Figure 3 (top-left) exhibits the exact same flaw, activating nonsensically on the image corners. This suggests the authors (or their LLM) are engaging in post-hoc rationalization, labeling the images (e.g., "black-and-white") and simply ignoring the contradictory heatmap evidence.
3. The evaluation is fatally flawed. It is circular: the method finds rare things, and the evaluation proves it found rare things. It is confounding: the method and its evaluation are critically dependent on external black-box models (CLIP, GPT-4o, LLaMA-4 Scout).
4. The method is brittle and relies on arbitrary thresholds, such as the "90th percentile" cutoff and the "0.003" cosine distance, with no sensitivity analysis or justification.

**Questions:**

1. In Figure 5, you dismiss low-scoring neurons because their heatmaps are "diffuse, fragmented, and fail to capture coherent semantic attributes". However, in Figure 3 (top-left, "Black-and-white photo..."), a high-scoring "minority" neuron clearly activates on the image corners, not the semantic content. How do you justify labeling this neuron as a coherent minority attribute while dismissing the ones in Figure 5 as noise? This appears to be a major contradiction.
2. The paper's premise links low activation frequency to "minority" or "suppressed" attributes. How do you distinguish between attributes that are "systematically suppressed" (i.e., the model learns to under-produce them relative to the training data) and attributes that are simply "naturally rare" (i.e., they have a low frequency in the training data)? Your method $s(z) = d \odot (1-\nu)$ appears to find both without distinction.
3. This is a critical point. Please clarify the exact training and inference pipeline. Do you train one global MSAE on a large, general-purpose corpus, or must you train a new MSAE for each prompt? What is the computational cost, and how does this scale as an auditing tool?
4. The semantic distinctiveness score $d_i$ is entirely dependent on CLIP. How can you be sure your method is not just discovering the biases of CLIP's embedding space rather than the diffusion model's internal representations?
5. The minority score $s(z) = d \odot (1-\nu)$ is a heuristic. Can you provide a stronger justification for this multiplicative form?

---

### Official Review · Reviewer_LaCj · 2025-11-01

**Soundness:** 2
**Presentation:** 3
**Contribution:** 2
**Rating:** 4
**Confidence:** 4

**Summary:**

This paper proposes MADGen, a post-hoc framework for discovering underrepresented “minority attributes” within text-to-image diffusion models by applying Matryoshka Sparse Autoencoders (MSAEs) to the internal bottleneck activations of the denoising UNet. The method ranks latent neurons based on a minority score combining activation rarity and semantic distinctiveness, and visualises them via top-activating samples and heatmaps. The authors show that MADGen can surface demographic, stylistic, and contextual biases across multiple Stable Diffusion variants, and demonstrate that prompting with discovered minority attributes can increase their presence in generated outputs. The framework is positioned as a representation-grounded auditing tool rather than a direct fairness mitigation method.

**Strengths:**

1. The paper addresses an important and timely problem: moving beyond predefined fairness axes to discover more general underrepresented attributes in diffusion models.
2. The use of MSAEs for hierarchical concept decomposition in diffusion activations is new in this specific context and provides a structured way to inspect internal model representations.
3. The experimental analysis includes cross-model comparison (SD v1.4, v2.1, SDXL) and explores multiple types of minority attributes (demographic, stylistic, contextual), demonstrating a broader scope than traditional fairness-only audits.

**Weaknesses:**

1. The technical contribution is limited: the method is largely a direct application of existing MSAE architectures to diffusion features, and the proposed “minority score” is heuristic rather than theoretically grounded.
2. Although the paper claims to be label-free, the pipeline critically depends on external vision-language models (GPT-4o) for semantic interpretation and attribute detection, meaning the attribute space is constrained by the biases and vocabulary of those models.
3. Although the method allows intervening on internal latent activations to increase the presence of minority attributes, this intervention still operates externally to the diffusion model and does not modify the model parameters or shift its intrinsic generative distribution.
4. The neuron visualisations are purely qualitative, and the paper does not provide quantitative validation that the discovered neurons are causally responsible for specific attributes. For example, there is no ablation, activation editing, or information-theoretic analysis to show that (i) neurons are disentangled, (ii) manipulating them reliably controls attributes, or (iii) they do not encode confounded signals.
5. Generalisation is not demonstrated: it is unclear whether MSAE-trained neurons transfer across prompts, attribute types, or datasets, and whether the identified attributes are stable or model-specific.

**Questions:**

1. How does MADGen compare to the baseline diffusion model in terms of inference or memory cost?
2. Can the authors demonstrate that the same MSAE-trained neurons remain meaningful across different prompts or datasets, rather than requiring per-prompt retraining?
3. The “amplification” experiment only edits prompts; have the authors evaluated whether the number of minority-attribute samples increases quantitatively (e.g., counting attribute occurrence before and after intervention)?
4. How robust is the method to the choice of external annotators (GPT-4o / Qwen-VL? If different models are used, do the discovered attributes change?

---

### Official Review · Reviewer_Z4F1 · 2025-11-07

**Soundness:** 3
**Presentation:** 4
**Contribution:** 3
**Rating:** 6
**Confidence:** 3

**Summary:**

This work addresses the issue of bias in text-to-image diffusion models, introducing MADGen, a framework designed to discover minority attributes in diffusion models, which uses Matryoshka Sparse Autoencoders and a novel minority metric to discover and amplify underrepresented attributes in diffusion models.

**Strengths:**

1. The paper is well-written and easy to follow. The figures are well-designed and enhance the understanding of the method.
2. This paper mentions a very promising issue in bias in text-to-image diffusion models. The framework is an effective method to systematically uncover latent minority attributes in diffusion models without predefined categories.

**Weaknesses:**

1. The experimental validation would benefit from broader comparative analysis. While the paper claims MADGen supports systematic auditing across Stable Diffusion variants (1.5, 2, and XL) and enables attribute amplification, the experimental results just employ Stable Diffusion v1.4 in quantitative comparison.

2. Quantitative comparison with state-of-the-art supervised debiasing methods[1-5] to clarify the trade-offs between unsupervised discovery and supervised correction would be interesting.

[1] Friedrich F, Schramowski P, Brack M, et al. Fair diffusion: Instructing text-to-image generation models on fairness[J]. arXiv preprint arXiv:2302.10893, 2023.

[2]Gandikota R, Orgad H, Belinkov Y, et al. Unified concept editing in diffusion models[C]//Proceedings of the IEEE/CVF Winter Conference on Applications of Computer Vision. 2024: 5111-5120.

[3]Shen X, Du C, Pang T, et al. Finetuning Text-to-Image Diffusion Models for Fairness[C]//The Twelfth International Conference on Learning Representations.

[4]Li J, Hu L, Zhang J, et al. Fair text-to-image diffusion via fair mapping[C]//Proceedings of the AAAI Conference on Artificial Intelligence. 2025, 39(25): 26256-26264.

[5]Li H, Shen C, Torr P, et al. Self-discovering interpretable diffusion latent directions for responsible text-to-image generation[C]//Proceedings of the IEEE/CVF Conference on Computer Vision and Pattern Recognition. 2024: 12006-12016.

**Questions:**

1. MADGen focuses on discovering minority attributes. However, if integrated with existing debiasing methods (such as prompt engineering), could potential conflicts arise? Can you provide more case and experiments demonstrating its compatibility with debiasing methods?
2. How does the method behave when dealing with multiple biases case, like both gender and race?
3. Whether using LLMs for attribute annotation may introduce extra bias due to their blind spots or inherent biases?
4. When using heatmaps to visualize interpretability results of generative images, did you explore the relationship between these interpretable features and conditional prompts?

---

### Note · Authors · 2025-11-13

I have read and agree with the venue's withdrawal policy on behalf of myself and my co-authors.